

# Interdecadal glacier inventories in the Karakoram since the 1990s

5  Fuming Xie[1,2], Shiyin Liu[1,2,3*], Yongpeng Gao[1,2], Yu Zhu[1,2], Tobias Bolch[4], Andreas Kääb[5], Shimei Duan[1,2], Wenfei Miao[1,2], Jianfang Kang[6,7], Yaonan Zhang[6,7], Xiran Pan[1,2], Caixia Qin[1,2], Kunpeng Wu[1,2], Miaomiao Qi[1,2], Xianhe Zhang[1,2], Ying Yi[1,2], Fengze Han[1,2], Xiaojun Yao[8], Qiao Liu[9], Xin Wang[10], Zongli Jiang[10], Donghui Shangguan[3], Yong Zhang[10], Richard Grünwald[2], Muhammad Adnan[1,2], Jyoti Karki[1,2], Muhammad Saifullah[11]

[1] Yunnan Key Laboratory of International Rivers and Transboundary Eco-Security, Yunnan University, Kunming 650500, China

[2] Institute of International Rivers and Eco-security, Yunnan University, Kunming, Yunnan 650500, China

[3] State Key Laboratory of Cryospheric Sciences, Northwest Institute of Eco-Environment and Resources, 15  Chinese Academy of Sciences, Lanzhou 730000, China

[4] School of Geography and Sustainable Development, University of St Andrews, St Andrews KY19 9AL, Scotland, United Kingdom

[5] Department of Geosciences, University of Oslo, Oslo, 0316, Norway

[6] Northwest Institute of Eco-Environment and Resources, Chinese Academy of Sciences, Lanzhou 20  730000, China

[7] National Cryosphere Desert Data Center, Lanzhou 730000, China

[8] College of Geography and Environmental Sciences, Northwest Normal University, Lanzhou 730070, China

[9] Institute of Mountain Hazards and Environment, Chinese Academy of Sciences, Chengdu 610041, 25  China

[10] School of Resource Environment and Safety Engineering, Hunan University of Science and Technology, Xiangtan, China

[11] Department of Agricultural Engineering, Muhammad Nawaz Shareef University of Agriculture, Multan, Pakistan

*Correspondence: Shiyin Liu (shiyin.liu@ynu.edu.cn)

## Abstract:

Multi-temporal glacier inventories provide key information about the glaciers, their characteristics and changes and are inevitable for glacier modelling and investigating geodetic mass changes. However, to 35  date, no consistent multi-tempo glacier inventory for the whole of the Karakoram exists, negatively affecting the monitoring of spatiotemporal variations of glaciers' geometric parameters and their related applications. We used a semi-automatic method combining automatic segmentation and manual



correction and produced multi-temporal Karakoram glacier inventories (KGI) compiled from Landsat TM/ETM+/OLI images for the 1990s, 2000s, 2010s, and 2020s. Our assessments using independent

multiple digitization of 37 glaciers show that the KGI is sufficiently accurate, with an overall uncertainty of ±3.68%. We also performed uncertainty evaluation for the contiguous glacier polygons using a buffer of half a pixel, which resulted in an average mapping uncertainty of ±3.68%. We calculated more than 20 attributes for each glacier, including coordinates, area, supraglacial debris area, date information, and topographic parameters derived from the ASTER GDEM. According to the KGI-2020, approximately

10500 alpine glaciers (> 0.01 km$^2$ each) cover an area of $22510 \pm 828$ km$^2$ of which $10.18 \pm 0.38\%$ ($2290 \pm 84$ km$^2$) are covered by supraglacial debris. Over the past three decades, the glaciers experienced a loss of clean ice/snow area but a gain in supraglacial debris. Supraglacial debris cover has increased by $17.63 \pm 1.44\%$ ($343.30 \pm 27.95$ km$^2$) while non-debris-covered glaciers decreased by $1.56 \pm 0.0.24\%$ ($319.85 \pm 49.92$ km$^2$). The total glacier area was relatively stable and showed only a slight insignificant increase

of $23.45 \pm 28.85$ km$^2$ ($0.10 \pm 0.13\%$). The glacier area has declined by $3.27 \pm 0.24\%$ in the eastern Karakoram while the glacier area slightly increased in central ($0.65 \pm 0.10\%$) and western Karakoram ($1.26 \pm 0.11\%$). Supraglacial debris has increased over whole Karakoram, especially in areas above 4200 m a.s.l., showing an upward shift. The glacier area changes were characterized by strong spatial heterogeneity, influenced by surging and advancing glaciers. However, due to global warming, the

glaciers are on average retreating. This is in particular true for small and debris-free glaciers. The multi-temporal KGI data are available at the National Cryosphere Desert Data Center of China: https://doi.org/10.12072/ncdc.glacier.db2386.2022 (Xie et al., 2022a)

## 1 Introduction

The Karakoram mountains are centred in northern Pakistan (see Fig. 1a and 1b) and host more than

20000 km$^2$ of glaciers, making this region one of the most glacierized areas outside of the polar regions. As the main source of the rivers' natural flow, meltwater from the Karakoram glaciers has a significant influence on the socio-economic development in the upper Indus River basin and the Tarim River basin areas to the north of the mountains (Winiger et al., 2005; Liu et al., 2006; Immerzeel et al., 2020; Liu et al., 2020; Nie et al., 2021). Due to global warming in recent decades, changes in glacier-related resources

and hazards present threats to the safety of critical infrastructure and the sustainable livelihoods of local communities living within the basin (Immerzeel et al., 2020; Bazai et al., 2021; Gao et al., 2021). In the past few decades, remote sensing-based mass balance, glacier flow, and surge or advance characteristics indicate that the glaciers in Karakoram are exhibiting anomalous behaviour (Hewitt, 2005; Gardelle et al., 2012; Kaab et al., 2012; Bolch et al., 2017; Dehecq et al., 2018; Farinotti et al., 2020; Wu et al., 2020;

Wu et al., 2021).

Glacier inventories provide fundamental baseline information about the glaciers and are needed for many applications (e.g. glacier mass balance, glacier modelling). Generally, a glacier inventory contains an identifier (ID), name (if available), location (i.e., coordinates), size, and other relevant information concerning each glacier (Paul et al., 2009; Paul, 2017). Since the 1970s, thanks to the development of

remote sensing and GIS technology, a series of international glacier inventories such as the World Glacier Inventory (WGI), the Randolph Glacier Inventory (RGI) or glacier inventories submitted to GLIMS



(Global Land Ice Measurements from Space) have been developed through the efforts of glaciologists from different countries (Müller and Scherler, 1980; Shih et al., 1980; Paul et al., 2002; Pfeffer et al., 2014). To date, several glacier inventories exist in the Karakoram area, some of which include information about supraglacial debris cover. These inventories include the Glacier Area Mapping for Discharge from the Asian Mountains (GAMDAM) glacier inventory (GGI15) (Nuimura et al., 2015) and its updated version (GGI18) (Sakai, 2019), the second Chinese glacier inventory (SCGI) (Guo et al., 2015; Liu et al., 2015), the International Centre for Integrated Mountain Development (ICIMOD) glacier inventory (Bajracharya and Shrestha, 2011), and the Glaciers_cci (CCI) glacier inventory (Mölg et al., 2018). However, due to differences and deficiencies in the remotely-sensed imagery, variable data years, and glacier outlines extraction methods used in previous glacier inventories, their results have poor comparability, thus failing to meet the requirements of glacier change analysis. Moreover, the presented areas of glacier coverage differ partially substantially for the different available inventories (Bolch, 2019; Bolch et al., 2019).

The two versions of the GAMDAM inventory (GGI15 and GGI18) were generated by manual delineation with the exclusion of glacierised areas in glaciated areas steeper than 40°. These inventories have a difference in an area of 9430 km$^2$. The differences can be mainly attributed to the omission of glacier areas above the bergschrund (Sakai, 2019). Moreover, the first one used Landsat images acquired around the year 2000 and the second one used also scenes spanning 1990~2010 in case the scenes used in the first version were unsuitable (Sakai, 2019). Whereas the ICIMOD inventory was compiled from Landsat TM/ETM+ images acquired between 2005 and 2009, the SCGI inventory only includes part of the Karakoram mountains and uses the Landsat TM/ETM+ satellite images between 2007 and 2011. The CCI inventory differs somewhat because the clean ice/snow parts are identified automatically using the image ratio technique and is based upon the Landsat images from 1998 to 2002. The debris-covered areas were manually mapped with the support of coherence images generated using ALOS-1 PALSAR-1 scenes from 2007 and 2009 (Mölg et al., 2018). In the RGI 6.0 (Rgiconsortium, 2017), the glaciers for the Karakoram region were extracted from the CCI inventory. A further approach to constraining changes in Karakoram glaciers was presented by Rankl et al. (2014), who focused only on surge-type glaciers and large glaciers with lengths > 3 km and areas > 0.15 km$^2$. Few studies present glacier area changes for smaller parts of the Karakoram (e.g. Minora et al. (2016) for the Central Karakoram National Park for 2001-2010 or Bhambri et al. (2013) for the Shyok Valley for 1973 to 2011). Both found overall no significant net are changes but large variability due to the presence of surging glaciers. However, consistent multi-temporal homogeneous glacier inventories covering the whole Karakoram mountains do not exist, thereby negatively affecting monitoring of spatiotemporal variations in the glaciers' geometric parameters and hampering explanation of their anomalous behaviour discussed by both international scientists and policymakers.

In most parts of the Karakoram, debris-covered glaciers are dominating (~10% of glacier area covered by debris). Thus, if this debris cover and its evolution are neglected, hydrological models may underestimate the longevity of glacier-sourced water resources and eventually overestimate the effects of glacier melt on sea-level rise (Herreid and Pellicciotti, 2020; Zhang and Liu, 2021). Many studies have reported that the supraglacial debris cover is expanding in different parts of the world (Mölg et al., 2019;





Tielidze et al., 2020; Xie et al., 2020b). Herreid et al. (2017) report no net change of the debris-covered area in Central Karakoram from 1977 to 2014. However, variations in the debris cover across the whole Karakoram region remains unknown. Mapping glaciers with debris cover is still challenging (Paul et al.,

2004; Pfeffer et al., 2014; Farinotti et al., 2020), with various data and approaches having been proposed to address this issue, including using SAR (Synthetic Aperture Radar) coherence maps, optical bands, thermal infrared features, terrain parameters, machine learning and other methods (Bhambri et al., 2011; Mölg et al., 2018; Alifu et al., 2020; Xie et al., 2020a).

The present study aims to present precise and homogeneous multi-temporal glacier inventories for

the whole Karakoram. We developed a high-performance cloud-removal and image-compositing algorithm to produce high-quality images for the glacierised regions of Karakoram for the extraction of glacier outlines. These will provide important baseline data for glacier research and related applications.

## 2 Data and methods

### 2.1 Landsat TM / ETM+ /OLI images pre-processing

At least 12 Landsat images are required to cover the glacierised area of the Karakoram (Fig. 1c). We selected Level 1 terrain-corrected products of the Landsat TM/ETM+/OLI scenes (L1TP) images with 30 m spatial resolution representing the years 1990~2020, including 65 TM, 39 ETM+ and 82 OLI images (Table S1). The data was identified and processed using Google Earth Engine. An Asia-North-Albers-Equal-Area-Conic projection coordinate system with the central meridian 76° and the standard

parallels 33° and 38° was used for all spatial data in this study. Firstly, the available images were initially filtered from the Landsat image collection by selecting the melting season (days of year: 200-270), region of interest (Karakoram boundary), and cloud cover (< 25~30%). The filter criteria depend on the data quality and observations recorded in different periods as shown in the lower-left corner of Fig. 1d. We set the image interval to two years to ensure representativeness and reliability of the composite image as

much as possible, a value smaller than the interval of three years deemed accepted in previous report (Paul et al., 2013). For the period of the 1990s, suitable images within one or two years were not sufficient to cover the whole Karakoram area, thus, two images from 1993 and 1994 were also selected for areas with no observations or unsuitable cloud conditions (see red box in Fig. 1d); this resulted in a more than three years interval in these areas, thus may increasing the inter-annual variation error of the glacier

outlines. Then, we applied a GEE's own cloud-removal algorithm (*ee.Algorithms.Landsat.simpleCloudScore*) (Gorelick et al., 2017) to all the images and composited a image needed for glacier inventory. In the image compositing, if more than 10 observations were reached at a given point, only the observations from the first 10 cloudiness images were accepted to generate an image composite in order to reduce the data source complexity of the composite image.

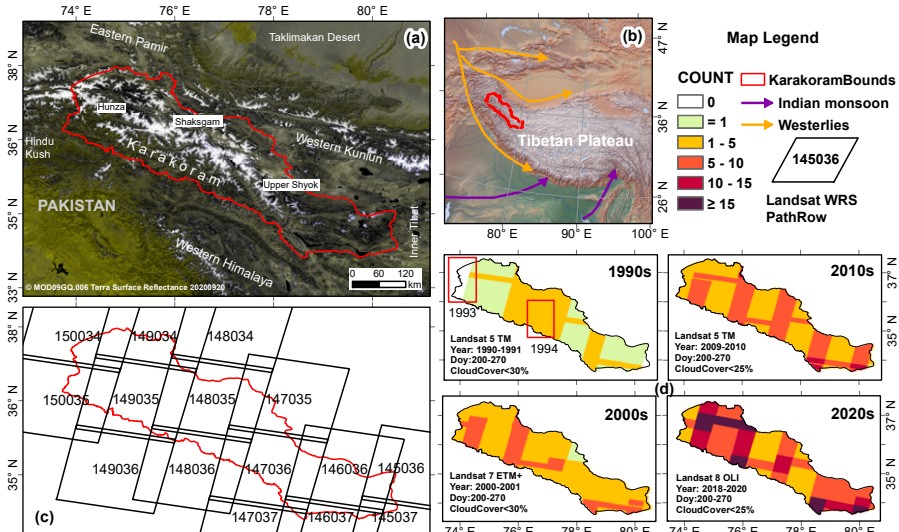

**Fig. 1 Overview of the study region (Karakoram Mountains). a)** Karakoram Mountains physiography. **b)** General climatology trends in Central Asia. The orange and blue arrow lines represent the westerly and monsoon directions, respectively (Yao et al., 2012). The Karakoram boundary was modified by reference to Bhambri et al. (2017). **c)** Worldwide reference system (WRS) Path/Row values of Landsat images used in this study. The blue area represents the glaciated area. **d)** Number of Landsat observations (COUNT) in each pixel, the details in the lower-left corner show the filtered image collection information: Landsat sensor name (TM/ETM+/OLI) and image acquisition time range, including year range, day of year (DOY) and cloud cover threshold. Due to the poor quality of satellite imagery in some areas (red rectangle) in 1990s, two scenes from 1993 and 1994 are used in the image composite.

### 2.2 Extraction of glacier outlines

Accurate glacier outline identification can be challenging, because glaciers exhibit a wide range of surface characteristics, they can be interpreted differently by the cartographer, surveyor, or remote sensing specialist (Paul, 2017; Paul et al., 2009). Moreover, poor quality satellite imagery (i.e., scenes with clouds, large shadows, or seasonal snow cover), debris-covered glaciers, rock glaciers, and perennial snow-covered depressions (potential presence of some ice underneath) are all sources of uncertainty in glacier outline mapping that may negatively affect the accuracy of the resulting outlines. Therefore we used a semi-automatic method (band ratio + manual correction) to compile glacier inventories. This approach is widely applied (Bolch et al., 2010a; Guo et al., 2015; Paul et al., 2017).

Firstly, we calculated the NDSI $(\rho_{green} - \rho_{swir1})/(\rho_{green} + \rho_{swir1})$ and initially considered areas with NDSI ≥ 0.4 as clean ice/ snow (or debris-free areas) (Dozier, 1989; Xie et al., 2020b) while other areas were labelled as initial debris-covered ice, followed manual correction (see next section). As previous studies have found, optimized thresholds for different periods can lead to overestimation of glacier area (Xie et al., 2020b); additionally, the glacier area has insensitivity to different thresholds when using high-quality satellite images on gentler terrain (Guo et al., 2015). Accordingly, we adopted the empirical NDSI threshold uniformly across the whole Karakoram for mapping clean ice/snow. A similar threshold was also used for generating glacier inventories for large regions elsewhere (e.g. Bolch et al.





(2010a)). Second, since many pixels outside of the glacier extent are considered to be supraglacial debris glacier outlines from previous glacier inventories were used as a mask, as suggested in similar studies (Bolch et al., 2010a; Scherler et al., 2018; Baumann et al., 2020). We combined two earlier glacier inventories (90% CCI + 10% GGI18) for KGI-1990s, and the subsequent KGI-2000s, KGI-2010s, and KGI-2020s rely on the earlier revision of the glacier inventory (hence, KGI-1990s, KGI-2000s and KGI-2010s). This approach is suitable with retreating glaciers but cuts off advancing glaciers. These were corrected by manual post-processing. Moreover, to remove the "salt-and-pepper" effect in the initial binary map, we applied a morphological filter with a $5 \times 5$ square kernel as described in Xie et al. (2020b). Then, the binary map was converted into vector format, retaining a 'Type' attribute describing whether each polygon is debris-covered ice or clean ice/snow, followed by the manual correction.

### 2.3 Manual correction

There are five typical error types in initial glacier outlines that require revision, including errors due to glacier omission (Fig. 2a), clouds or shadow (Fig. 2b), glacier commission (Fig. 2c), glacier advance or surge (Fig. 2g), and errors due to debris and ponds. To eliminate these errors, we have established detailed criteria and applied manual correction of multi-temporal glacier outlines using Landsat SWIR-NIR-RED or NIR-RED-GRE (false) colour composites of images (30 m resolution). These were combined with other optional baseline data, such as the 15 m-resolution panchromatic Landsat band, the land surface temperature (LST) map derived from Landsat thermal infrared band (Kraaijenbrink et al., 2017; Liao et al., 2020), contour lines derived from ASTER GDEM and Google Earth images. Most of the debris-covered ice sections were remapped using the panchromatic band, LST, topography, and Google Earth images, while the ice tongues of debris-covered glaciers were identified by the location of their meltwater outlets (see Fig. 2 (e) and (f)). The emergence of ice cliffs and ponds on glacier surfaces was also used in our present glacier inventory to identify the site of debris-covered glaciers and classify them as part of the debris-covered ice. Paul et al. (2013) observed that there are large uncertainties in manually identified glacier outlines, due to the influence of seasonal snow cover, shadows and complex underlying glacier surface types, such as supraglacial debris cover, rock glaciers, stagnant ice, etc., resulting in large deviations of the glacier extent even by different experienced interpreters. This issue is especially prevalent for glacier tongues with thick debris, which completely depends on visual interpretation to map the glacier, thus leading to further uncertainties. Therefore, to reduce the artificial interferences caused by subjectivity, all the correction work in this study was performed by the main author only.

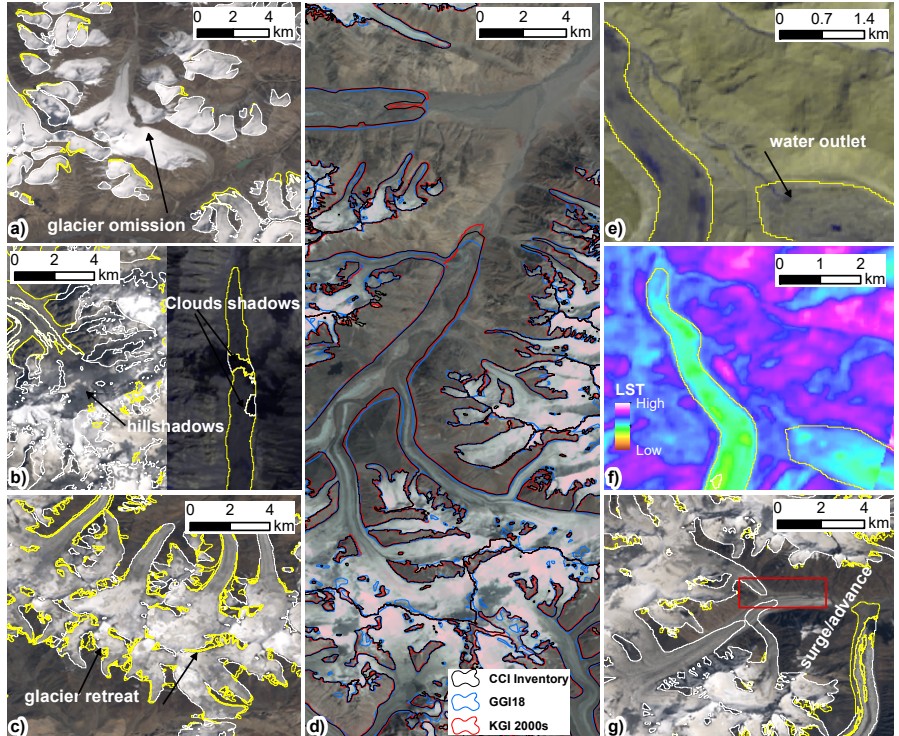

**Fig. 2 Types of manual correction. a)** Glaciers missed in previous inventories. **b)** Areas contaminated by hill shadows or cloud shadows. **c)** Glaciers that have retreated and require updating. **d)** Comparison of the revised glacier outlines with those from the CCI and GGI 18 glacier inventories, with a base map of Landsat 7 ETM+ composited in the 2000s. **e)** and **f)** are auxiliary methods for mapping debris-covered glaciers. The location of the water outlet is used to determine the position of the glacier tongue, and the land surface temperature images are used to distinguish between debris-covered glaciers and exposed bedrock. **g)** Glacier terminus advance may-be caused by glacier surges (red rectangle). Except for subfigure d, the yellow and white lines represent the glaciers identified by the automatic threshold segmentation method as debris-covered and debris-free parts, respectively.

### 2.4 Ice divides

Segmentation of complex glacier polygons, such as glaciers connected in their accumulation areas, can be achieved by using ridgelines derived from a digital elevation model (DEM) (Mölg et al., 2018). Thus, we adopted the method proposed by Bolch et al. (2010a), using the ASTER GDEM v3 dataset. Nuimura et al. (2015) used the ICESate GLA 14 data to evaluate DEM accuracy data in glaciers of high mountain Asia and found that the ASTER GDEM version 2 (v2) is more suitable in comparison to the SRTM DEM. Compared to ASTER GDEM v2, the updated v3 product has less voids areas due to an increased number of ASTER stereo image data and further developed processing that improves the vertical and horizontal accuracy of the data (Abrams, 2016). Therefore, topographic information for all glaciers in this study was obtained from ASTER GDEM v3 data. First, we merged the four periods of glacier outlines into a new version with a 1 km buffer. This was then used to clip the sink-filled DEM. A flow direction grid was then calculated to generate a raw drainage basin using watershed analysis (Fig.





3). Then manual corrections were performed to delete small basins (<0.01km²), based on a three-dimensional topographic map derived from the DEM datasets and a hill-shadow (see the base map of Fig.3). The ridgelines were then extracted to split glacier complexes into single glaciers. After identifying the ice divides, we performed some batch processing on the revised glacier outlines, including filling voids (eliminating contained parts only) with an area of less than 4000 m² (i.e., less than 5 pixels), deleting small supraglacial debris patches (< 4000 m²) and eliminating glaciers smaller than 0.01km². This resulted in glacier area changes of only 1.63 to max. 6.97 km² in the different periods.

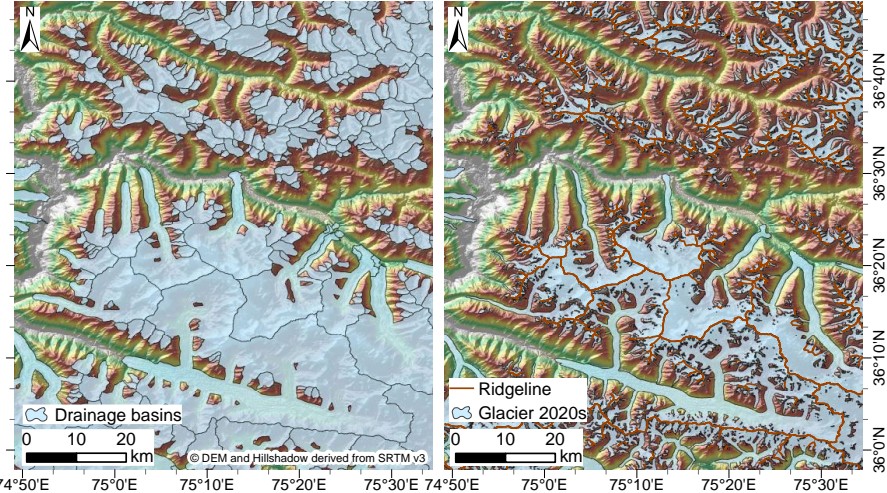

**Fig. 3** Glacier drainage basins derived from ASTER GDEM v3 and ridgelines extracted after manual correction.

2.5 Attribute data

We included more than 20 attributes for each analysed glacier. In addition to the GLIMS ID linked from the RGI inventory, each glacier was given a unique KGI ID according to the GLIMS ID encoding style that is automatically calculated via Python programming using the centroid latitude and longitude of the glacier polygon. For terrain attributes, the minimum, median, and maximum elevation, plus median slope and aspect were calculated from the ASTER GDEM v3. Since in particular large glaciers (> 5 km²) may consist of different branches and tributary glaciers with different aspects or glaciers have large bends, the resulting representativeness of the average aspect may be poor; we thus also appended an attribute of a manually identified aspect according to the approximate orientation of each glacier. In addition, other information, such as longitude, latitude, glacier area, debris area, name, date, glacier surface type (clean ice or debris), terminus status (whether advancing), as well as the name of the operator were included. For the corresponding date of all inventoried glaciers, we provide note only the year range of the composited image dates but also calculate the probability that the edge pixels of the glaciers originate from a certain date. This is on the premise that the interannual difference of pixels inside the glacier does not affect the accuracy of glacier extent. However, this assumption may affect the supraglacial debris cover. As shown in Fig. S1, we generated a date image to track the source image for each pixel in the composited image; this indicated that the composite images had little interannual variation for most glaciers in the inventory as the pixels in most composite scenes were derived from data from the same



255    year.

### 2.6 Estimation of the uncertainty

Potential error-sources of glacier inventory include errors from satellite data quality (i.e., due to seasonal snow cover), DEMs, methods of glacier delineation, complex underlying glacier surface types, and quality of ridgelines (for single glaciers). In general, we applied two different uncertainty assessment

methods. The first method evaluates the glacier outlines derived from coarser spatial resolution satellite imagery using the glacier outlines identified from high resolution images or mean value of multiple digitisations from lower-resolution data. In an ideal case the outlines should be digitized several times (multi-manual digitization (MMD)) to consider the variability of the analysts ("round robin") (Paul et al., 2013). In the "round robin" method, seven experienced participants repeatedly digitized 37 glaciers

between 0.10 km$^2$ and 100 km$^2$ in size, located in the Hunza valley, western Karakoram based on high-resolution images (Sentinel-2 and Planet) and lower-resolution images (Landsat 8), the spatial locations are shown in Fig. S2. The standard deviation (STD) of the digitized glacier area (MMD results) values is used to evaluate the precision of the analysts' digitisation and the difference between its mean area value (as the "truth" of glacier area) and the area calculated from the glacier inventory are used to

determine which glacier extent has lower uncertainty, i.e., the KGI or the MMD outlines. Results of MMD are shown in Table S2. The digitization on the Landsat OLI scene had STD of 1.34-25.93% (mean 2.29%), with the largest value for the small glacier and partly debris-covered glaciers (Fig. 4), while the STD of on Sentinel-2 scene was smaller with a mean of 1.28% (from 0.48% to 7.39%). Apart from errors at debris-covered tongues or margins, interpretation differences in the accumulation area are considered

a key error source, including differences in interpretation of snow patches and glacier extents (Fig. 4). Additionally, scatter plots (Fig. S3) of the linear regression for both the MMD and KGI-2020s glacier area showed a strong positive correlation between them, with a determination coefficient (R$^2$) value of 0.99 and a root mean square error (RMSE) of 0.78 km$^2$, thus confirming the high accuracy of the KGI data. The mean area difference of KGI outlines and the MMD mean values based on the Landsat image

is -6.92%; 5.58% based on Sentinel-2 images. The mean area difference between the KGI outlines and the high-resolution image with only one digitization is -6.39%(Sentinel-2) and 2.56%(Planet). The area difference for the different glaciers varied from to -124.73% to 35.24% (-1.90 ~ 2.61 km$^2$), influenced by glacier size, presence of tributary glaciers, and whether glaciers were debris-covered or debris free. The resultant total mapping uncertainty is ±3.68%, which meets the requirements of the Global

Observing System for Climate (Gcos, 2016). The results presented above are consistent with previous accuracy assessments that reported the mean area differences between the automatically derived and manually digitized outlines on high-resolution images between 2% and 5% (Andreassen et al., 2008; Paul et al., 2013; Paul et al., 2017).

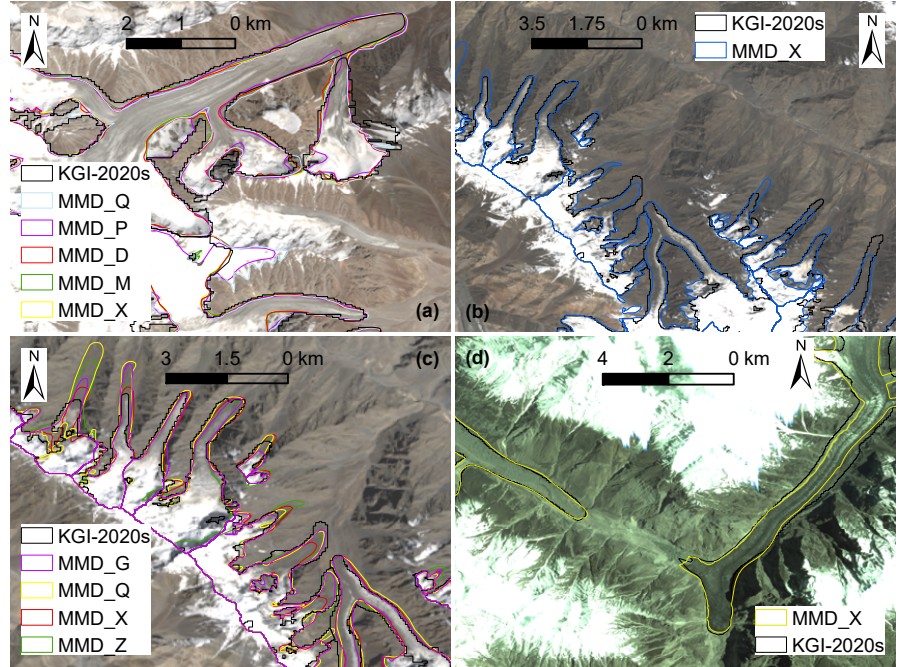

**Fig.4** Glacier extent digitized by different participants based on Sentinel-2 image (**a**, **b**), Landsat image (**c**) and Planet image (**d**) with KGI-2020s derived from Landsat 30-m imagery.

As a second measure of uncertainty, we applied the buffer method (Bolch et al., 2010a; Granshaw and G. Fountain, 2017) on the contiguous glacier polygons. Accordingly, a buffer of ± 1/2 pixel (i.e., 15 m) for the KGI outlines were generated and the difference of the area of the buffer and the KGI was used as the uncertainty measure. The difference for the four periods are ± 5.31%, ± 5.18%, ± 5.12% and ±5.21%, with an average difference of ±5.21%. In terms of the debris-covered areas, generally, a buffer of ± 1 or 2 pixels (30 or 60 m) buffer was suggested in previous research (Mölg et al., 2018; Paul et al., 2020). The uncertainty of the debris portion in this study was evaluated through the ratio of the glacier area to the debris cover area multiplied by the uncertainty of ± 1 pixel (30m) buffer, resulting in uncertainty of ±26.9%, ±28.9%, ±27.6%, and ±29.2% for the four KGI periods, with a mean value of ± 28.1%. For the whole glacier, the mapping uncertainty based on buffer distance is higher than estimated based on the "round robin" experiment indicating that the 'round robin' value is a good estimation of the uncertainty. Hence, we used this value ($\sigma = \pm\ 3.68\%$) as the uncertainty value for all KGI data in this study. The area change uncertainty ($\sigma_\Delta$) was estimated according to the standard error propagation, as root sum square of the uncertainty for outlines mapped from different periods, but only consider the glacier parts which showed change in the 1990s and 2020s ($\Delta A_{1990s}$ and $\Delta A_{2020s}$) (Bhambri et al., 2011; Zhang et al., 2018; Li et al., 2022), calculated as : $\sigma_\Delta = \sqrt{(\Delta A_{1990s} * \sigma)^2 + (\Delta A_{2020s} * \sigma)^2}$.

## 3 Results

### 3.1 Characteristics and status of the Karakoram glaciers

We mapped a total of 10,498 glaciers with an area of 22510.73 ± 828.39 km$^2$ (Table 1). All glaciers are divided into 9 areal classes of ≤0.05 km$^2$ to ≥100 km$^2$. 7941 (75.60% of all glaciers) are smaller than 1 km$^2$, but cover only a total area of 2098.45 ± 77.22 km$^2$ (9.30% of the total). 26 glaciers exceed 100 km$^2$ which covers 7430.64 ± 273.45 km$^2$ or 32.5% of the total glacier area in Karakoram. Siachen (1118.09 ± 41.14 km$^2$), Baltoro (849.44 ± 31.26 km$^2$), Biafo (579.25 ± 21.32 km$^2$) and Hispar (542.07 ±

19.95 km$^2$) glaciers are larger than 500 km$^2$. 20% (2175) of the glaciers with an area within size classes 1~5 km$^2$ and 10~50 km$^2$ cover a total area of 8519.63 ± 313.52 km$^2$. Glaciers with size classes 5~10 km$^2$ and 50~100 km$^2$ are roughly equal in area, each class covering about 2,200 km$^2$. 581 glaciers, ranging from 5 to 100 km$^2$ cover 8736.78 ± 321.51 km$^2$, meaning that all these glaciers plus the 26 glaciers exceeding 100 km$^2$ occupy 72% of the total area of the Karakoram glaciers.

We divided the Karakoram into northern and southern slopes (NK and SK) by ridgelines, and into eastern, central, and western parts (EK, CK and WK) using the Indus and Tarim sub-basin outlines (Fig.5). 55.3% of the glaciers are concentrated in CK, covering an area of 12547.12 ± 461.73 km$^2$, followed by the glaciers in the WK. Less than one-fifth, i.e., 3954.30 ± 145.52 km$^2$ or 17.60 ± 0.89% of the total glaciers are located in EK. A high proportion of debris cover is a typical feature of Karakoram glaciers,

with a debris-cover percentage of 2.63 ± 0.10% (104.94 ± 3.83 km$^2$) in the eastern and 11.11 ± 0.41% (1382.68 ± 50.88 km$^2$) and 13.16 ± 0.48% (804.32 ± 29.60 km$^2$) in the central and western regions, respectively. Additionally, the glacier areas in NK and SK differ by about 20%, 12321.74 ± 453.44 km$^2$ and 10188.99 ± 374.95 km$^2$. However, the supraglacial debris cover in SK (1500.85 ± 55.23 km$^2$, 65.51 ± 2.41%) is around twice than recorded in NK (790.09 ± 29.08 km$^2$).

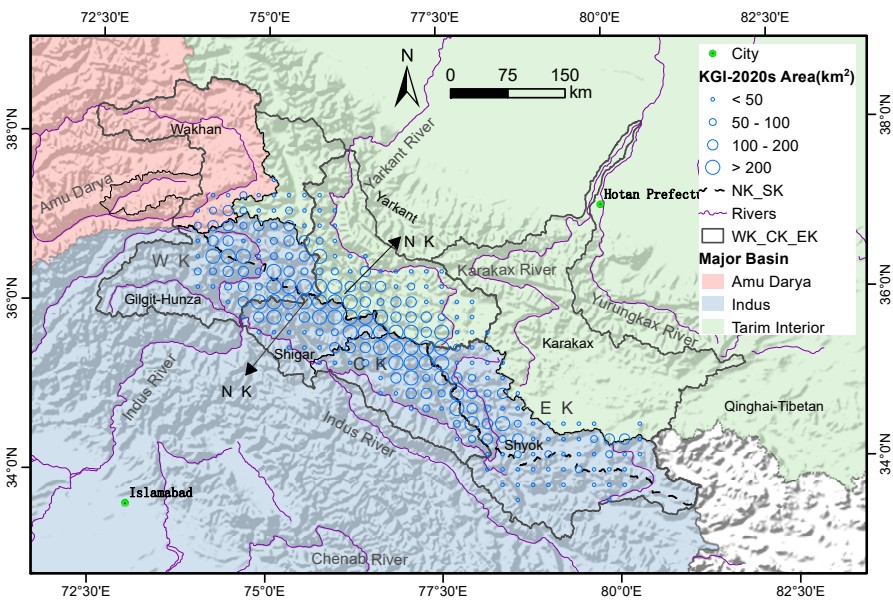



**Fig. 5**. The Karakoram Mountains were divided into western, central and eastern Karakoram (WK, CK and EK) according to the Indus and Tarim sub-basins, and into Northern and Southern Karakoram based on the main central ridgeline. The major sub-basin divisions in the Karakoram mountains also are shown. Hollow circles represent the glacier areas, as aggregated on a 20 km × 20 km grid.

Moreover, we divide the Karakoram mountains into five sub-basins: Gilgit-Hunza, Shigar, Shyok, Wakhan and sub-Tarim (Yarkant, Karakax and Qinghai-Tibetan) (Fig. 5). Among these, the Shyok sub-basin housed the largest extent of glaciers (7918.54 ± 291.40 km$^2$), followed by the Tarim sub-basin (5574.06 ± 205.13 km$^2$), then Gilgit-Hunza (4974.34 ± 183.06km$^2$), and Shigar (3481.02 ± 128.10 km$^2$). The percentage of debris cover is greatest in the Shigar (16.64%) and Gilgit-Hunza (14.72%) basins,

while sub-Tarim basin has the lowest debris coverage (5.87%).

The total glacier area in the Karakoram was overall relatively stable from 1990 to 2020, with a slight, insignificant increase of 23.45 ± 28.85 km$^2$ (0.10 ± 0.13%) on average. However, the total glacier number decreased by 1. 36% in this period. Small glaciers (< 1 km$^2$) experienced a strong decrease in area, particularly those smaller than 0.05 km$^2$. In contrast, large glaciers with areas exceeding 100 km$^2$ showed

an increase in area, which compensated for the total area decrease for smaller glaciers to some extent. In terms of time series, the total glacier area showed an increasing trend from 1990 to 2010, while a decreasing trend after 2010s. This may indicate a weakening of the abnormal behaviour of glaciers in the Karakoram owing to the continuous warming.

### 3.2 Supraglacial debris cover of Karakoram glaciers

Supraglacial debris cover is widespread in the Karakoram. 1848 debris-covered glaciers (17.63% of all glaciers) with a total area of 2290.95 ± 84.31 km$^2$ of supraglacial debris area (~10.18 ± 0.38% of total glacier area) were mapped in the KGI-2020s. The total area of debris-covered glaciers is more than 16800 km$^2$, of which 74.4% of the glaciers (1374) are larger than 1 km$^2$. The supraglacial debris area of glaciers smaller than 1 km$^2$ is only 49.73 ± 1.83 km$^2$, accounting for 2.17 ± 0.08% of the total debris coverage.

Hence, there is no substantial debris cover on these small glaciers. The debris cover area of the 26 glaciers larger than 100 km$^2$ dominates, representing almost half (47.5%) of the total debris cover area with an area of 1088.83 ± 40.07 km$^2$. This is followed by the 10~50 km$^2$ size class glaciers, which cover a total area of 413.90 ± 15.23 km$^2$ (18.10 ± 0.67%). Also, despite the glacier number being on the same order of magnitude in size classes 5 ~10 km$^2$ and 50 ~100 km$^2$, the latter's debris coverage (321.21 ± 11.82

km$^2$) is twice than of the former (170.76 ± 6.28 km$^2$), indicating that the larger the glacier, the greater the likelihood and severity of its surface debris cover.

From 1990 to 2020, the debris cover has increased by 17.63 ± 1.44% (343.30 ± 27.95 km$^2$), compared to a 1.56 ± 0.24% decrease (319.85 ± 49.92 km$^2$) in the clean ice or snow part. All different size classes show debris cover increases on their surfaces (Table S3). Debris cover area on glaciers > 100

km$^2$ and in size classes in 1~5 km$^2$ present the most significant changes, increasing by 113.18 ± 10.87 km$^2$ (11.60 ± 1.11%) and 79.12 ± 5.08km$^2$ (44.16 ± 2.84%) respectively. A similar trend was also found in glaciers smaller than 1 km$^2$, in which the debris cover area increased by 19.47 ± 1.32 km$^2$ (64.34 ± 4.36%) in total; however, this accounted reached for only 5.67% of the total recorded debris increase. Except for glaciers smaller than 1 km$^2$, the growth rates of debris coverage in size classes between 1~5

km$^2$ (~ 44.16 ± 2.84%) and 5-10 km$^2$ (32.14 ± 2.12%) are the most significant. Our results indicate that



supraglacial debris cover is expanding on large debris-covered glaciers, while many small and medium-sized debris-free glaciers are transforming into debris-covered glaciers. Over the study period, approximately 500 debris-free glaciers developed into debris-covered glaciers with a decadal growth rate of 12.20% (i.e., about 166 transforms into debris-covered glaciers every decade), more than 97% of which

are smaller than 10 km$^2$. According to KGI-2020s, among the glaciers larger than 5 km$^2$, there are 457 (14379.96 ± 529.18 km$^2$) glaciers with supraglacial debris coverage greater than 0.1 km$^2$. The area of these debris-covered glaciers increased by 98.06 ± 13.43 km$^2$ (0.69 ± 0.09%) from 1990 to 2020, while that of debris-free glaciers decreased by 12.72 ±2.64 km$^2$ (0.68 ± 0.14%).

### 3.3 Glacier elevation, slope and aspect

The Karakoram glaciers have a wide range of elevations with a mean altitude of ~ 5300 m a.s.l. and a median altitude of ~5500 m a.s.l. (5200 ~ 6200 m a.s.l.). The hypsometry of the glacier area (Fig. 6a and 6b) indicates that the distribution of glaciers is concentrated at the altitude 5300~5700 m a.s.l., whereas the debris-covered sections are primarily distributed at altitudes of 4000~5000 m a.s.l. and gradually disappear at altitudes above 6,000 m a.s.l.. The altitudinal profiles of the glacier surface area

(Fig. 6b) indicate that glaciers lose area at low altitude, while near the altitude zone at 5000 ~ 6000 m a.s.l, it shows an increasing trend during 1990-2010, followed a decreasing trend since the 2010s. In terms of glacier orientation (Fig. 6c and 6d), most glaciers have a south, east or southeast aspect (~ 52.40%), with only a few glaciers facing to the north. While there are equal numbers of east and southeast-facing glaciers, the area of the southeast-facing glaciers is twice as large as that of the east-

facing glaciers. Similarly, the area of the north-facing glaciers is not as small as that of the west-facing glaciers, which occupy the smallest area. The southeast and north-facing glaciers in the Karakoram mountains are dominated by large glaciers. As depicted in Fig. 6e and 6f, the slope of all glaciers is mainly concentrated between 0 ~ 50°, with almost no glaciers with slopes greater than 70°, whereas supraglacial debris is widespread in areas with slopes less than 15°. This may be due to broad, flat valley

areas providing favourable terrain conditions for the accumulation and enrichment of supraglacial debris. Previous studies have used a slope value of less than 24° (Paul et al., 2004) or 25° (Xie et al., 2020b) to limit the extent of debris, even though there are steep slopes with debris, which may result from steep ice cliffs or crevasses. Overall, compared to debris-free glaciers, debris-covered glaciers have a broader altitude range (i.e., lower minimum elevation and higher maximum elevation) and lower slopes (Table

S4).





**Table 1.** Number and area of glaciers according to different size classes or surface types during 1990~2020.

| Glacier Size (km²) / Surface type | Glacier Number (GN) | | | | Glacier Area (GA) km² | | | | GN (%) 2020s | GA (%) 2020s | GN Change NC(%) (1990~2020) | GA Change AC(%) (1990~2020) |
|---|---|---|---|---|---|---|---|---|---|---|---|---|
| | 1990s | 2000s | 2010s | 2020s | 1990s | 2000s | 2010s | 2020s | | | | |
| <0.05 | 1120 | 1007 | 843 | 1096 | 40.14 ± 1.48 | 33.84 ± 1.25 | 29.53 ± 1.08 | 35.93 ± 1.32 | 10.44 | 0.16 | -2.14 | -10.49 |
| 0.05 ~ 0.1 | 1723 | 1659 | 1622 | 1654 | 124.89 ± 4.60 | 119.98 ± 4.42 | 117.39 ± 4.41 | 119.9 ± 4.41 | 15.76 | 0.53 | -4 | -4 |
| 0.1 ~ 0.5 | 3798 | 3764 | 3732 | 3752 | 936.61 ± 34.47 | 933.35 ± 34.34 | 928.24 ± 34.16 | 924.21 ± 34.01 | 35.74 | 4.11 | -1.21 | -1.32 |
| 0.5 ~ 1 | 1440 | 1459 | 1466 | 1439 | 1022.48 ± 37.63 | 1038.84 ± 38.23 | 1042.74 ± 38.37 | 1018.41 ± 37.48 | 13.71 | 4.52 | -0.07 | -0.40 |
| 1 ~ 5 | 1951 | 1945 | 1971 | 1950 | 4213.42 ± 155.05 | 4223.14 ± 155.41 | 4279.34 ± 155.47 | 4214.86 ± 155.11 | 18.57 | 18.72 | -0.05 | 0.03 |
| 5 ~ 10 | 319 | 331 | 327 | 326 | 2199.79 ± 80.95 | 2294.42 ± 84.43 | 2271.23 ± 83.58 | 2231.65 ± 82.12 | 3.11 | 9.91 | 2.19 | 1.45 |
| 10 ~ 50 | 238 | 231 | 234 | 225 | 4549.64 ± 167.43 | 4419.86 ± 162.54 | 4462.6 ± 164.22 | 4304.77 ± 158.42 | 2.14 | 19.12 | -5.46 | -5.38 |
| 50 ~ 100 | 30 | 29 | 30 | 30 | 2259.27 ± 83.14 | 2149.36 ± 79.10 | 2223.37 ± 81.82 | 2200.36 ± 80.97 | 0.29 | 9.77 | 0 | -2.61 |
| >100 | 24 | 26 | 26 | 26 | 7141.04 ± 262.79 | 7364.39 ± 271.01 | 7450.43 ± 274.18 | 7430.64 ± 273.44 | 0.25 | 33.01 | 8.33 | 4.06 |
| Debris-free | 9291 | 8914 | 8773 | 8650 | 20539.63 ± 755.86 | 20414.01 ± 751.24 | 20789.49 ± 765.05 | 20219.78 ± 744.09 | 82.40 | 89.82 | -6.90 | -1.56 |
| Debris-covered* | 1074 | 1199 | 1229 | 1374 | 1947.65 ± 71.67 | 2160.26 ± 79.50 | 2015.37 ± 74.17 | 2290.95 ± 84.31 | 13.08 | 10.18 | 27.93 | 17.63 |
| Total | 10643 | 10451 | 10251 | 10498 | 22487.28 ± 827.53 | 22574.27 ± 830.73 | 22804.86 ± 839.22 | 22510.73 ± 828.39 | 100 | 100 | -1.36 | 0.10 |

* Glacier Number: number of debris-covered glaciers greater than 1 km²; Glacier Area: total area of all debris-covered ice



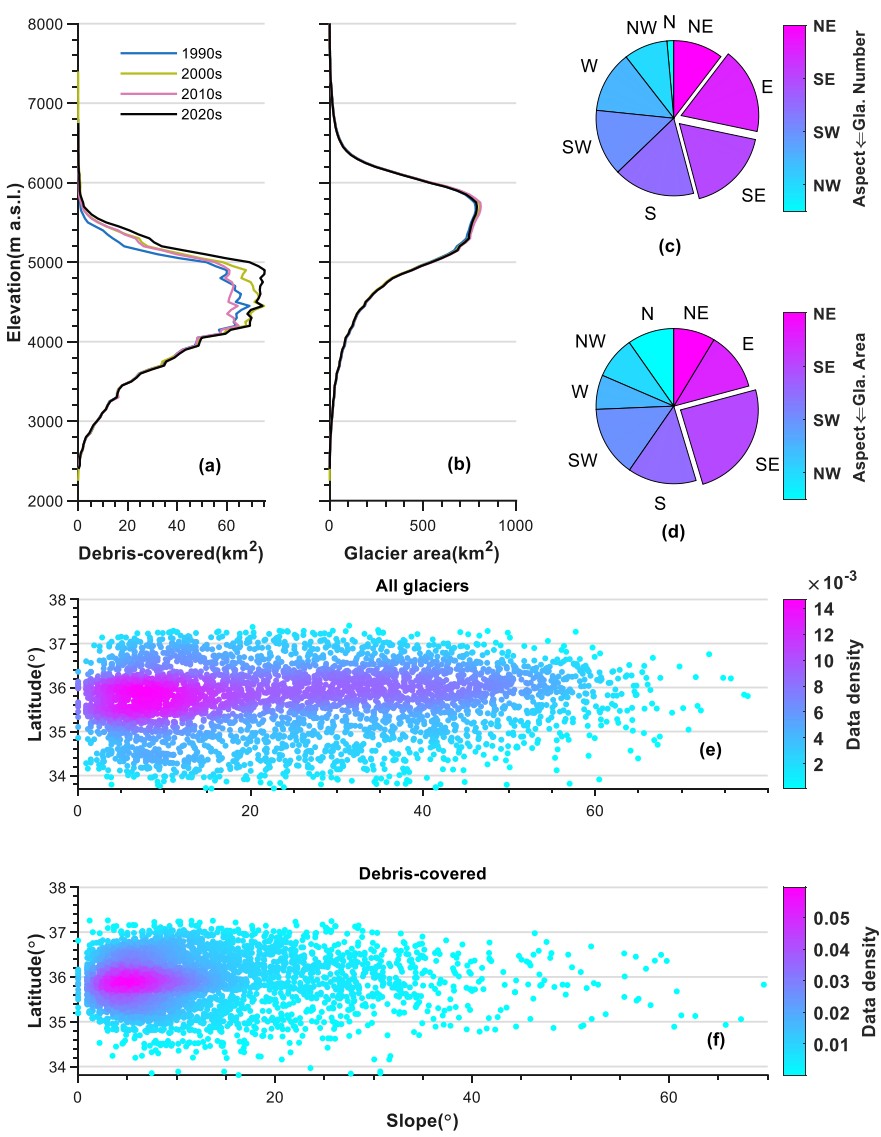


**Fig.6** Altitudinal profiles of the glacier surface area at 50 m intervals for debris-covered ice **(a)** and all glaciers **(b)**, showing variations from 1990 to 2020. Plots showing glaciers' number and area by aspect **(c and d)**, in addition to surface slope versus latitude for glaciers and debris-covered sections (**e** and **f**).

## 4 Discussion

### 4.1 Comparison with previous glacier inventories

In order to determine major differences between previous glacier inventories and our glacier inventory in the Karakoram, we compared it with the SCGI (5114.68 km$^2$, glacier area within the KGI range), ICIMOD (11612.82 km$^2$), CCI (20444.40 km$^2$), and GGI18 (20036.55 km$^2$) inventories. The





Karakoram boundary used by us is a little different from that in previous studies (Bolch et al., 2019; Bolch et al., 2012), so the comparison is only for areas covered by both inventories. Among these, the SCGI and ICIMOD inventory were compared with the KGI-2010s. We then contrasted the GGI18 and CCI inventory with the KGI-2000s to minimize the inter-annual variation between the compared data. For comparison the glacier areas were aggregated onto a 5 km × 5 km grid (Fig. 7). Our results indicate that there is a substantial difference between the ICIMOD inventory and KGI data. In some areas, the glacier area in a single grid cell is underestimated by 15.38 km$^2$ (253.40%), with a total RMSE of 3.78 km$^2$. The GGI18 and SCGI inventories exhibited the differences from the KGI data between 1.45~7.11 km$^2$ and 1.34~8.60 km$^2$, with RMSE values of 1.41 km$^2$ and 1.53 km$^2$, respectively. The difference between the CCI inventory and KGI-2000s was found to be the smallest, with RMSE and R$^2$ values of 0.65 km$^2$ and ~1 (0.997), while the difference in glacier area per grid was between 2.42~3.32 km$^2$. The clear underestimation of glacier area in the ICIMOD inventory may relate to the fact that clean ice or snow cover in high altitudes, steep headwalls, and areas affected by hill shadows or clouds were not considered (Bajracharya and Shrestha, 2011) (see also Fig. S4). Similar factors still affect the updated GAMDAM inventory (GGI18), resulting in glacier area values smaller than ours in most regions; in contrast, the difference between the CCI and KGI inventories was relatively small because a similar approach was adopted, with only differences in data source and interpreter (see Fig. 2d). In addition to the above factors, different interpreters' identification of debris-covered glaciers and inter-annual variation between comparative data are potential error sources in all glacier inventories. To sum up, such glacier inventories produced through different methodological standards, data sources, and interpreters contain many uncertainties from different sources, and therefore homogeneous multi-temporal glacier inventories are required to calculate glacier changes.

We performed another comparison in regions where both previous data and our inventory mapped supraglacial debris. The results indicate that the debris cover area in the CCI inventory is higher (18.47 km$^2$) than that of KGI-2000s, while the total debris cover mapped by (Scherler et al., 2018) is much lower than KGI-2020s (in total, ~18 % less). The latter is a particularly large difference, which may be related to the inter-annual variation (five years), lack of checks or manual correction, and overlooking the terminal changes caused by glacier surges or advances in the results extracted by Scherler et al. (2018). Moreover, the supraglacial debris cover on glaciers larger than 2 km$^2$ mapped by Herreid and Pellicciotti (2020) using Landsat images from 1986 to 2016 (median 2013) was 2095.96 km$^2$, while that of area for glaciers in four periods of KGI were 1859.24 km$^2$, 2045.21 km$^2$, 1931.95 km$^2$, and 2163.05 km$^2$ (mean 1999.86 km$^2$, ~5% less). Hence, compared to previous glacier inventory or supraglacial debris data, our KGI data has obvious advantages in aspects including Landsat image compositing, glacier initial outline mapping, and manual correction. Thus, the quality of KGI data is considered to be reliable and sufficiently accurate.

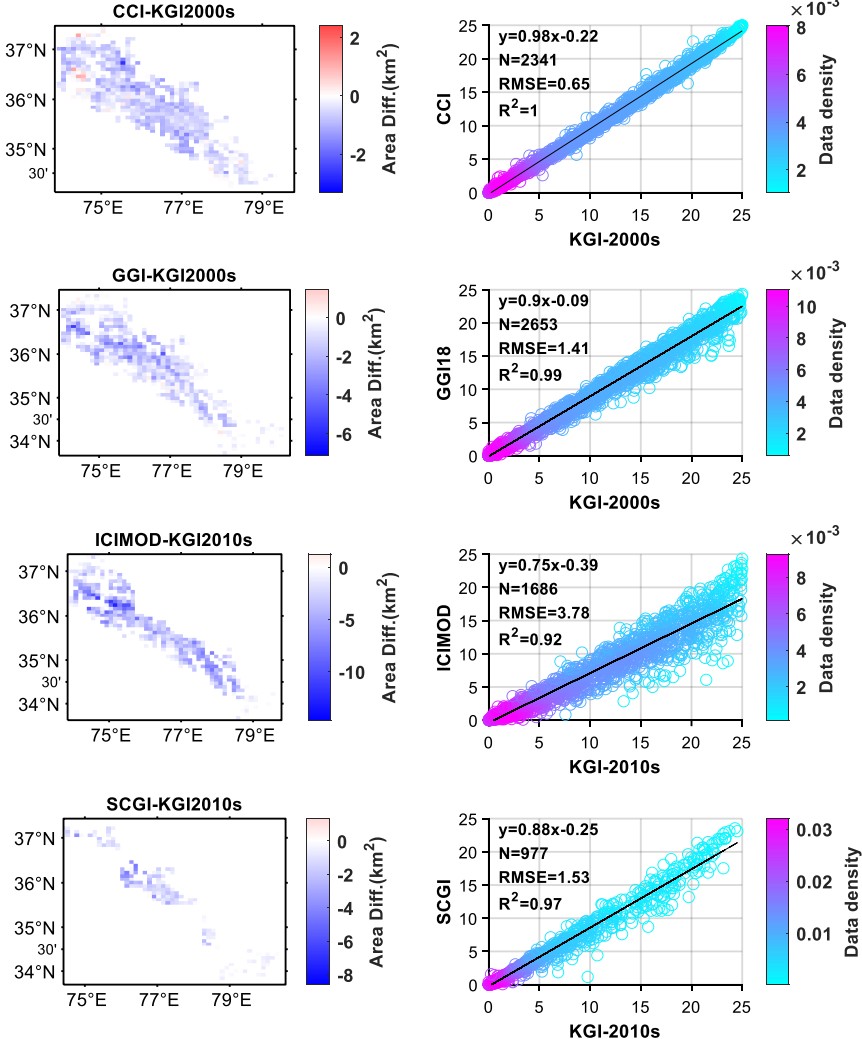

Fig.7. Comparison between the KGI inventory and previous glacier inventories. All data are aggregated on a 5km × 5km grid. The map in the left column represents the area difference between previous glacier inventories and the KGI inventory from the same or closest period (SCGI and ICIMOD inventories are compared with KGI-2010s, while the updated GAMDAM (GGI18) and CCI inventories are compared with KGI-2000s). The right column illustrates, from top to bottom, the scattergrams of glacier area in each 5 km grid cell of the KGI inventory, plotted against the GGI18, CCI, ICIMOD and SCGI inventories in the Karakoram mountains.

**4.2 Spatial-temporal variations of the Karakoram glaciers**

Our results indicate that the glacier area has declined by $3.27 \pm 0.24\%$ ($133.50 \pm 9.84$ km$^2$) in EK, in contrast to a slightly increased area in CK ($0.65 \pm 0.10\%$, $80.89 \pm 12.61$ km$^2$) and WK ($1.26 \pm 0.11\%$, $76.06 \pm 6.49$ km$^2$) (Fig. 8, Table S5). This finding is also consistent with the negative mass balance for EK glaciers in the past decade and balanced budgets in other parts of the Karakoram (Vijay and Braun,



2018; Berthier and Brun, 2019). However, contrary to the area changes, the number of glaciers in CK and WK has decreased by 2.82% and 2.24%, respectively. Nevertheless, due to the influence of seasonal or annual variations in snow patches resulting in uncertainties in glacier number changes, especially for small glaciers, these changes might not be significant. Furthermore, our results show that glaciers in the northern slope of the Karakoram mountains are potentially undergoing a loss in area (-0.49 ± 0.08%), while the area of southern slope glaciers has potentially increased by 0.84 ± 0.18%. Hence, it is of particular importance to consider topographic conditions when explaining of spatial heterogeneity in glacier change as also shown elsewhere (Xie and Liu, 2009; Garg et al., 2017). We suggest that this may be due to the wetter conditions, as well as the increased precipitation the south side of the main divide, allowing glaciers in this area to continue to grow and store (Xie and Liu, 2009; Dimri, 2021).

The abundance of supraglacial debris may also partially explain the variability in glacier change observed in the Karakoram. The debris cover increased across the whole region, with EK experiencing the fastest growth (an increase of 105.05± 5.16% or 53.25 ± 2.62 km$^2$) during the past three decades. This trend was followed by CK (18.99 ± 1.47%, 220.65 ± 17.09 km$^2$) and WK (9.44±1.13%, 69.40 ± 8.27 km$^2$). The debris cover in both NK and SK areas has increased by roughly the same order of magnitude (~170 km$^2$). On the southern slope of the Karakoram, 13.81 ± 0.51% of the glacierized areas are covered by debris, and the glacier area increased by 0.84 ± 0.19%, while on the northern slope, the glacier area shrank by 0.49 ± 0.08% under a relatively small debris coverage (5.60 ± 0.21%). These results show that there is a relationship between debris coverage and glacier area change, i.e., the area of glaciers with higher debris coverage is increasing or remaining stable and vice versa.

From 1990 to 2020, the debris coverage has notably increased in areas above 4,200 m a.s.l., indicating that supraglacial debris in the Karakoram was experiencing the same upward evolution. This was also observed in other mountains such as the Greater Caucasus (Tielidze et al., 2020) and western Swiss Alps (Mölg et al., 2019), and is an inevitable evolution process of debris-covered glaciers under global warming (Herreid and Pellicciotti, 2020). Increased snow avalanche activity at high altitudes may have brought more debris to the glacier (Hewitt, 2005), thus leading to an increase in supraglacial debris, in addition to the effects of global warming, such as thinning of the glaciers and expose more englacial debris, and increased numbers of paraglacial slope failures (Zhang et al., 2011; Zhong et al., 2022; Zhang et al., 2015).

By focusing on terminus changes, we observed that the advance exceeds the recession for most of the glaciers. Similar to Rankl et al. (2014), we identified 65 terminus-advancing glaciers with a total area of 1,647.19 ± 60.62 km$^2$. In the KGI-2020s, we found 63 single glacier units, since two of the glaciers have already become branches of other existing glaciers over the past three decades, all of which greatly contributed to the increase of glacier area by up to 25.33 ± 0.93 km$^2$. Fig. 8 also shows a clear spatial coincidence between glacier regions with increased area and distribution of the advancing glaciers, which presumably could also be confirmed by the amount of increased area and terminal change area of the advanced glaciers. In addition, our data shows that most advancing glaciers (~70%) are surge-type glaciers, which represents abnormal glacier behaviour in the Karakoram compared to other regions in high mountain Asia (Farinotti et al., 2020). The most recent inventory of surge-type glaciers (Guillet et al., 2022) has complied 223 glaciers in the Karakoram. The number of which is consistent with the results

of Bhambri et al. (2017), but the glaciers were not completely identical. In KGI-2020s, there are 607 glaciers in size classes greater than 5 km$^2$, including 157 surge-type glaciers, which have expanded by 89.92 ± 6.97 km$^2$ (0.89 ± 0.07%) in total area and 136.03 ± 14.73 km$^2$ (10.61 ± 1.15%) in supraglacial debris coverage in the past three decades; while the 450 non-surge-type glaciers lost area of 4.58 ± 6.61

505   km$^2$ (0.08 ± 0.11%). Compared to normal glaciers, these anomalous glaciers (i.e., surging and advancing glaciers) are directly responsible for the area growth of the Karakoram glaciers.

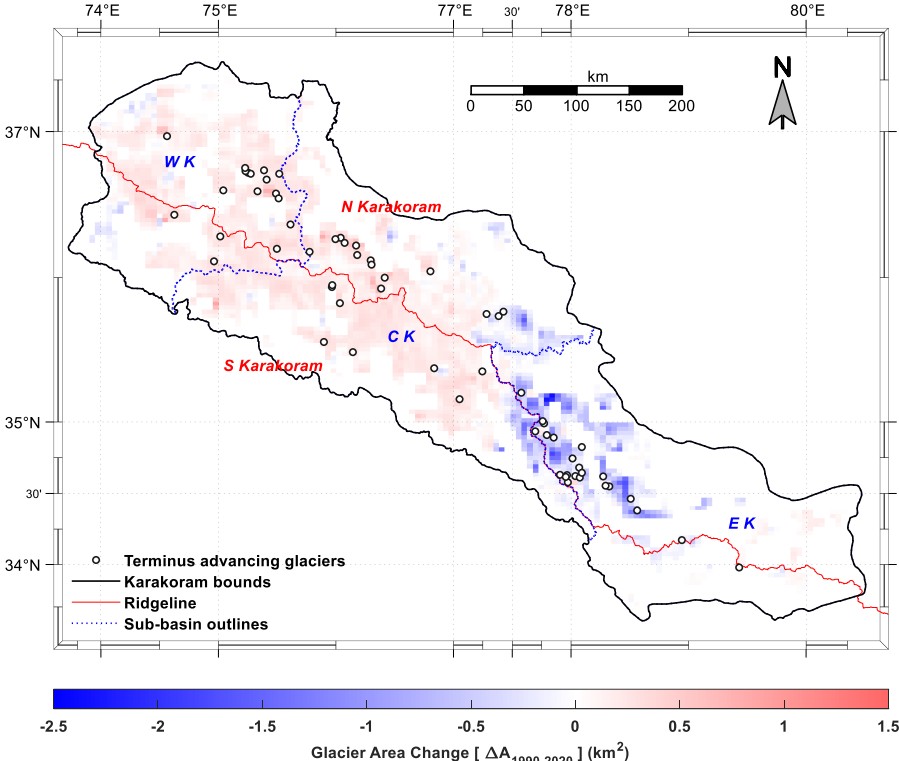

**Fig.8 Glacier change in the Karakoram during 1990~2020**. Glacier area aggregated on a 20 km × 20 km grid. Hollow dots represent the glaciers advanced during the period 1990-2020. The Karakoram Mountains were divided

into western, central and eastern Karakoram (WK, CK and EK) according to the Indus and Tarim sub-basins , and into Northern and Southern Karakoram based on the main central ridgeline.

### 4.3 Application of the multi-temporal glacier inventories

In the melt-dominated Tarim and Indus basins, accelerated glacier melt is the main contributor to rising 21st century streamflow, which increases before peak water, then declines (Huss and Hock, 2018;

Rounce et al., 2020; Nie et al., 2021). When computing glacier mass changes and associated runoff from projection models, multi-temporal glacier inventories are an important data source for either basic glacier outlies or as a validation set. Based on published ice thickness data, we calculated the ice volume in Karakoram is 2.03 ± 0.52 ×10$^3$ km$^3$ (Farinotti et al., 2019) or 2.81 ± 1.08 ×10$^3$ (Millan et al., 2022), which has a potential contribution to sea-level rise of 4.88 ± 1.27 mm or 7.11 ± 3.07 mm. Taking into

account different glacier extents in different periods, these projections will produce a variable error of

0.36 ~ 0.49%. Moreover, referring to the suggestions of Braithwaite and Raper (2009) and (Sakai et al., 2015), we assume that median glacier altitude could act as a proxy for long-term ELA, which is correlated with the glacier mass balance budget. The glacier ELA is a sensitive indicator of climate changes that a small rise in air temperature can result in a larger uplift (Shi and Liu, 2000; Liu et al., 2014), that can be used to describe the state and fate of glaciers (Fujita and Nuimura, 2011). Among the five sub-basins, as shown in Table S6, the median altitudes of glaciers in the three basins with increasing glacier coverage decreased, while the altitudes increased in the basins with decreasing glacier area. Spatially, as pointed out by Bolch et al. (2012), the median elevations increase with the distance from the moisture source (Fig. S5). The glaciers in the northwest exposed to the westerlies and heavily debris-covered have a relatively low median elevation, while the glaciers north or northeast of the main ridge of the Karakoram have a clearly higher median elevation. On the whole, the median elevation of the Karakoram glaciers showed an increasing trend during 1990-2020, indicating that glacier melting likely is becoming more intense, with runoff moving towards peak water (Huss and Hock, 2018; Nie et al., 2021).

**4.4 Challenges in multi-temporal glacier inventories**

Algorithms for identifying debris-covered glaciers are have been developed but the accuracy is too low if no manual corrections area applied. Like the method in the present study, semi-automated approaches are the most prevalent, which will always rely on existing glacier boundaries or time-consuming manual corrections (Paul et al., 2004; Bolch et al., 2007; Bhambri et al., 2011; Racoviteanu and Williams, 2012; Smith et al., 2015; Mölg et al., 2018). Although pixel-based machine learning has been used to map debris-covered ice (Alifu et al., 2020; Xie et al., 2020b; Xie et al., 2022b), isolated pixels, irregular outlines, or voids in the results have forced it to be revised to meet the needs of glacier inventories. In addition, available remotely sensed imagery with high quality (cloud-free, low or no seasonal snow, small shadows, geometrically calibrated, etc.) is required for precisely compiling glacier inventories. Even if satellite images from the end of the ablation    season are used, some mapping errors may complicate the results for investigating glacier changes, due to interannual variations in snow cover in the melting season or remaining seasonal snowfields (Yi et al., 2021). For example, during the 2010s when the overall glacier area was abnormal and the area of clean ice/snow part was high, the identified debris cover was significantly lower as a result of the high snow cover in 2009 (see Fig. S6).

To obtain inventories of the time before higher resolution multi-spectral earth-observation satellites were launched (e.g. Landsat 4, launched in 1982), topographic maps were often used to map glacier outlines. However, these have large uncertainties (Bhambri and Bolch, 2009; Bolch et al., 2010b; Guo et al., 2015; Paul, 2017). For the 1960s and 1970s, declassified Corona KH-4 (spatial resolution 8-2m)/Hexagon KH-9 (12-8m) data, and Landsat MSS (60 m) images were used to compile inventories in some regions (Bolch et al., 2008; Bolch et al., 2010b; Schmidt and Nüsser, 2012; Bhambri et al., 2013; Gardent et al., 2014; Pieczonka and Bolch, 2015; Weber et al., 2020; Bhattacharya et al., 2021). However, the available optical imagery used in current glacier inventories is often limited due to occlusion by frequent cloud layers during the melting season; this, in turn, increases the interannual variation in the inventory results and further prolongs the time interval between multitemporal glacier inventories. Fortunately, this situation improved with the launch of Sentinel-2A and B and Landsat 8 and 9, except in



cloudy areas, like south-eastern Tibetan Plateau. Although SAR coherence maps have been used in previous glacier inventories to overcome these limitations in optical imagery, most focus on the identification of supraglacial debris (Frey et al., 2012; Robson et al., 2015; Ke et al., 2016; Mölg et al., 2018; Alifu et al., 2020) and only a few studies have been used to map glacier outlines (Atwood et al., 2014; Yang et al., 2016; Lippl et al., 2018). Therefore, in the future, greater availability of cloud

computing platforms to develop SAR image processing algorithms and make extensive use of optical remote imagery will greatly help the development of multitemporal glacier inventories. Moreover, if glacier extents in past periods can be reconstructed based on the characteristic   glacial landform like lateral and terminal moraines (Shi and Liu, 2000) using existing aerial imagery and satellite maps, the temporal resolution of multi-temporal glacier inventories can be extended, such as for glacier inventories

from the Little Ice Age (e.g. Lucchesi et al. (2014)and Meier et al. (2018)).

## 5 Conclusions and outlook

        Glacier inventories are the baseline information for many applications such as hydrological modelling climate change studies. In this study, first we generated inventories which allowed us to systematically detect glacier change patterns in the Karakoram range over the past three decades by using

186 Landsat scenes through a semi-automatic method based on the Google Earth Engine cloud-based platform. Validation using the "round robin" method showed an overall mapping uncertainty of ± 3.68% which is within the accuracy range required for glacier inventories by GCOS. We also compared the KGI with previous glacier inventories and evaluated it by applying uncertainties of ± 1/2 pixel (15 m) and 1 pixel (30 m) buffers to the glacier outlines. These assessments indicate that our results are reliable and

sufficiently accurate. In the 2020s, there were approximately 10500 glaciers in the Karakoram mountains, covering an area of $22510.73 \pm 828.39$ km$^2$ of which $10.18 \pm 0.38\%$ ($2290.95 \pm 84.31$ km$^2$) is covered by supraglacial debris. During the past 30 years, the glaciers experienced a loss of clean ice area and a gain in supraglacial debris. The total glacier area in the Karakoram remained relatively stable from 1990 to 2020, with a slight, but insignificant increase of $23.45 \pm 28.85$ km$^2$ ($0.10 \pm 0.13\%$), while the

supraglacial debris cover increased clearly by $17.63 \pm 1.44\%$ ($343.30 \pm 27.95$ km$^2$), in contrast to a $1.56 \pm 0.24\%$ ($319.85 \pm 49.92$ km$^2$) decrease in clean ice/snow. These glacier areal changes were characterized by strong spatial heterogeneity and dominated by surging and advancing glaciers. However, the glaciers are experiencing the negative effects of faster snow/ice melting caused by global warming, with small and clean glaciers being more sensitive. This poses a growing threat for regional water scarcity and

glacier-related hazards in the downstream basin. Most importantly, the KGI data in this study will provide basic glacier outlines for GLIMS and the forthcoming planned release of the next versions of the RGI, as well as for other glacier-related research applications.

## 6 Data availability

        The KGI datasets are available from National Cryosphere Desert Data Center of China at

https://doi.org/10.12072/ncdc.glacier.db2386.2022 (Xie et al., 2022a) for users' assessments and applications. The data files contain shapefiles for the complete glaciers and the debris-covered section. Additionally, a description file is included.

        **Author contributions.** SL designed the framework of glacier inventory. FX programmed the glacier



mapping algorithms, manually revised glacier outlines and ridgelines, and wrote the first manuscript. SL
        edited the first draft. TB reviewed and improved the manuscript. FX, YZ, YG, SD, WM, FH, XR and CQ
        digitized glaciers using Sentinel-2 imagery. All authors discussed and improved the manuscript.

        **Acknowledgments.** We thank the Google Earth Engine developer for the freely available cloud-
computing platform and USGS/NASA for Landsat imagery, ASTER GDEM version 3 product. We
        acknowledge previous glacier inventory data from Glaciers_cci, GAMDAM glacier inventory, ICIMOD
        glacier inventory as well as the Second Chinese Glacier Inventory for supporting this study results.

        **Financial support.** This research has been supported by the International Science and Technology
Innovation Cooperation Program of the State Key Research and Development Program (Grant No.
        2021YFE0116800), the National Natural Science Foundation of China (Grant No. 42171129), the Open
        Foundation from National Cryosphere Desert Data Center(Grant No.2021KF01), the Second Tibetan
        Plateau Scientific Expedition and Research program (Grant No. 2019QZKK0208), the Director Fund of
        the International Research Center of Big Data for Sustainable Development Goals (Grant No.
CBAS2022DF020), the Postgraduate Research and Innovation Foundation of Yunnan University (Grant
        No. 2021Z018) and the Scientific Research Fund project of Yunnan Education Department (Grant
        No.2022Y059).

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
