# Peer review of "Interdecadal glacier inventories in the Karakoram since the 1990s"

_Earth System Science Data, 2022_

## Author Comment (AC1)

On behalf of all the co-authors, I would like to thank the reviewer, Referee #2, Rakesh Bhambri, for his thoughtful and constructive comments which helped us to improve our study. We have responded to comments as follows:

**NOTE**
Anonymous Referee #1 Comments (Black font)
Authors Responses (Red font)
*Specific changes that were made in the manuscript (Blue italic)*

1. This study presented a new glacier inventory for four time periods (1990, 2000, 2010, 2020) covering the Karakoram and surrounding region (upper Shyok basin) using Landsat satellite imagery and reported insignificant area loss in the study area. The manuscript is very well-written and nicely structured. I have given some minor suggestions for improvement. The important issue is an outline of the Karakoram region. The present study modified the extent of Karakoram (L131) presented by Bhambri et al. (2017) but did not mention the reasons for this change. Bhambri et al. (2022) recently reported no international standardization on the Karakoram extent. Therefore, consistency in the spatial extent of the Karakoram region is needed to quantify, analyze, and compare databases of natural and cultural resources for scientific investigation on a common platform and harmonization of scientific studies. Comparing glacier numbers and area statistics with previous studies is impractical (section 4.1) as all the studies on Karakoram glaciers have different area coverage. Bhambri et al. (2022) provided a most appropriate digital outline of the Karakoram region based on two decades (1920s and 1930s) long discussions and descriptive enumerations of the Royal Geographical Society (RGS) and the Survey of India (SoI). I suggest using this most common outline (open access) for the extent of the Karakoram and using the same outline to extract previous glacier inventory data on the same platform for comparison and modify section 4.1. If you do not want to use this outline, for the sake of harmonizing scientific studies, you can change the title to "Interdecadal glacier inventories in the Karakoram and the surrounding region since the 1990s".

Response: Thank you for your valuable comments. We noticed that there are several boundaries in the Karakoram Mountains with subtle differences in extent. The Karakoram boundary we used is a revised version with reference to Bhambri et al. (2017) developed by our team (can be freely accessed with the link "https://github.com/1923xfmingynu/Subdivision-Of-High-mountain-Asia") (see Figure 1).

In section 4.1, we make comparison between our glacier inventory with others at the same region scales, for instance, when we compared with SCGI, we identified the minimum part based on glaciers in SCGI contained in our Karakoram boundary to ensure the comparison has practical significance. We hold that the original title "Interdecadal glacier inventories in the Karakoram since the 1990s" is more reasonable.

[Figure]

Figure 1 Sub-divisions of high mountain Asia

Suggestions

2. L41-42 Uncertainty is the same ±3.68 in two sentences. If it is the same, write accordingly. You can write ±3.7.

Response: Thank you for pointing out the mistake. Actually, it's different. We corrected it.

"*Our assessments using independent multiple digitization of 37 glaciers show that the KGI is sufficiently accurate, with an overall uncertainty of ±3.68%. We also performed uncertainty evaluation for the contiguous glacier polygons using a buffer of half a pixel, which resulted in an average mapping uncertainty of ±5.21%.*"

3. L50 Present results in single-digit after the point (23.4 ± 28.8 km2). Please carefully check the entire manuscript. In some places, it is single-digit (e.g., L300), and in others in double-digit.

Response: Thank you for your suggestion. We have modified the results in the whole paper to retain the two-digit decimal accuracy.

4. L59 Most glaciological studies usually avoid referring to countries' names for the Karakoram region. If you mention Pakistan in the first sentence of the introduction, then India and China must also be mentioned for the sake of neutrality. If you like, you can refer to the contested nature of this particular mountain region with different territorial claims between the different nation-states in a very general way. This is one aspect which creates continuous problems for ground truthing and field measurements. See Baghel and Nüsser (2015).

Response: Thank you for your suggestion. We have revised it to avoid referring to countries' names for the Karakoram region.

"*The Karakoram mountains are centred in **western Tibetan Plateau** (see Fig. 1a and 1b) and host more than 20000 km² of glaciers, making this region one of the most glacierized areas outside of the polar regions.*"

5.  L89 "Moreover, the presented areas of glacier coverage differ partially substantially for the different available inventories (Bolch, 2019; Bolch et al., 2019)." Here you can mention Bhambri et al. (2022).

Response: Thank you. We have updated the corresponding references.

6.  L91 "delineation with the exclusion of glacierized areas in glaciated areas steeper than 40°." Here two terms, ' glacierized' and 'glaciated', are used, and I could not understand them. Please see Cogley et al. (2010) for these terms.

Response: Thank you for your suggestion. We revised this sentence and unified the use of professional terms.

"*The two versions of the GAMDAM inventory (GGI15 and GGI18) were generated by manual delineation with the exclusion of **glacierised areas steeper than 40°.***"

7.  L133 "The data were identified and processed using Google Earth Engine." For image processing or glacier mapping?

Response: This refers to image processing. We have modified this sentence. The preliminary extraction of glacier outlines is implemented on GEE, while the manual revision and statistical analysis are finished on the local computer.

"*The satellite images were identified and processed using Google Earth Engine (GEE).*"

8.  L153 for Karakoram boundary modified…. Please see my comment above.

Response: Referring to the reply to the first comment, we changed the statement to "*The Karakoram boundary is a reasonable revised boundary with reference to Bhambri et al. (2017) and can be accessed freely via "https://github.com/1923xfmingynu/Subdivision-Of-High-mountain-Asia*". And we will add boundary data to the data assets.

9.  L175 Bolch et al. (2010) used TM3/TM5 band ratio instead of NDSI. Therefore, Bolch et al. (2010) TM3/TM5 band ratio threshold must be different from NDSI.

Response: Thank you for your suggestion. We corrected the references.

"*A similar threshold was also used for generating glacier inventories for large regions elsewhere (e.g. **Ke et al. (2016)**)*"

10. L201 Please omit etc.

Response: Thank you for your suggestion. "etc." has been removed

11. L218 Double space between can be

Response: Thanks. The error has been corrected.

12. L223 Double space between developed processing

Response: Thanks. The error has been corrected.

13. L293 This paper was published in 2006 (Granshaw and G. Fountain, 2017). Please check.

Response: Thank you for pointing out the mistake. The corresponding references have been updated and the references in the entire manuscript have been checked.

14. L394 between 0 "and" 50°

Response: Thank you, it has been revised.

15. L414 "Karakoram boundary used by us is a little different from that in previous studies (Bolch et al., 2019; Bolch et al., 2012)," I don't think this is little difference. Also, please see my suggestions for the Karakoram boundary above.

Response: As stated in the reply to the first comment. Here we have made further modifications and clarifications.

*"However, due to the different approaches, data sources and methods among different glacier inventories, cannot be compared without a high level of uncertainty, so this is only a qualitative comparison. The Karakoram boundary used by us is different from that in previous studies (Bolch et al., 2019; Bolch et al., 2012; Bhambri et al., 2022), so the qualitative comparison is also only for areas covered by both inventories."*

16. L438 Scherler et al., (2018)

Response: Thank you. We have corrected the citation format of the references.

References

Baghel, R. and Nüsser, M., 2015. Securing the heights: The vertical dimension of the Siachen conflict between India and Pakistan in the Eastern Karakoram. Political Geography 48, pp. 24-36.

Bhambri R., Chand, P., Nüsser, M., Kawishwar, P., Kumar, A., Gupta, A.K., Verma, A., Tiwari, S.K., 2022. Reassessing the Karakoram Through Historical Archives - Environmental Change in South Asia: Essays in Honor of Mohammed Taher, in: Saikia, A., Thapa, P. (Eds.),. Springer International Publishing, Cham, pp. 139–169. https://doi.org/10.1007/978-3-030-47660-1_8

Bolch, T., Menounos, B. and Wheate, R., 2010. Landsat-based inventory of glaciers in western Canada, 1985–2005. Remote sensing of Environment, 114(1), pp.127-137.

Cogley, J.G., Arendt, A.A., Bauder, A., Braithwaite, R.J., Hock, R., Jansson, P., Kaser, G., Moller, M., Nicholson, L., Rasmussen, L.A. and Zemp, M., 2010. Glossary of glacier mass balance and related terms.

Granshaw, F.D. and Fountain, A.G., 2006. Glacier change (1958–1998) in the north Cascades national park complex, Washington, USA. Journal of Glaciology, 52(177), pp.251-256.

---

## Author Comment (AC2)

On behalf of all the co-authors, I would like to thank the reviewer, Anonymous Referee #1, for his thoughtful and constructive comments which helped us to improve our study. We have responded to comments as follows:

**NOTE**

Anonymous Referee #1 Comments (Black font)

Authors Responses (Red font)

*Specific changes that were made in the manuscript (Blue italic)*

1. The manuscript deals with the compilation of a glacier inventory in the Karakoram region. The topic is of high interest to the scientific community, and not only. The manuscript is well written (there are a few typos to check, e.g. line 248), but there are some issues to be solved before its publication. First of all, authors need to describe all the, employed, data at the beginning of Section2; currently, ancillary data, which constitute an important part of the processed ones, are progressively introduced during the description of the various elaboration.

Response: Thank you for your valuable suggestions. We have made modifications according to your suggestions, including adding a section (Sec. 2.1) to introduce all the data used in this study, and described the data in detail in Table 1.

"*2.1 Datasets* **(Partial content)**

*This subsection lists all data sets covering the Karakoram that are used to produce and assist in the analysis of the multi-temporal glacier inventory, including optical images from different satellite sensor, digital elevation model (DEM), four previous glacier inventories, three supraglacial debris extents, two surge-type glacier inventories, two modelled ice thickness data, hydrological basins and river networks. Table 1 summarizes their key characteristics, presenting their sources, date, application in this study and access link.*

*At least 12 Landsat images are required ……*

*Table 1 Lists of data sets covering the Karakoram mountain that are used in this study. (* **Partial content**)

| Data Name | Sources | Date | Access |
|-----------|---------|------|--------|
| *Satellite images* | *Landsat TM, ETM and OLI +30-m/ 15-m images* | *1990, 1991, 1993, 1994; 2000, 2001; 2009, 2010; 2018, 2019, 2020 (details see Table S1)* | *GEE asset or https://earthexplorer.usgs.gov/* |
| | *Sentinel-2 10-m images* | *2020-08-25, 2020-08-23* | *GEE asset or https://scihub.copernicus.eu/* |
| | *Planet 3-m images* | *2019-05-29* | *Ordered and download via Planet's APIs* |
| *DEM* | *30-m ASTER GDEM V3* | *2000-2013* | *https://e4ftl01.cr.usgs.gov/ASTT/ASTGTM.003* |
| … | … | … | … |

"

2. Concerning subsection 2.6, to improve readers' comprehension, a figure, like Fig. 4, should be added. Moreover, if applied to debris-covered glaciers, the discrepancies between the two methods highlight their proneness to errors in their mapping.

Response: Thank you for your suggestions. We added a figure (Fig. 5) and rephrased the text.

Among the feasible methods to determine accuracy and precision of glacier outlines(Paul et al., 2017), the buffer method (most used) and multiple digitizing (including area difference) are the most commonly used and effective methods (Mölg et al., 2018; Paul et al., 2020; Paul et al., 2017; Guo et al., 2015)(Mölg et al., 2018; Paul et al., 2020; Paul et al., 2017; Guo et al., 2015). The buffer method provides a minimum/maximum estimate of precision that scales with glacier size. Its overall value will thus vary with the size distribution of the selected sample. Due to the mapping uncertainty of the debris-covered glaciers is usually greater than clean ice, a 30m buffer was used to evaluate the uncertainty of the debris-covered part in this study, as in previous studies, it is also treated differently (e.g., ±2% for clean ice, ±5% for debris-covered, or±5%) (Paul et al., 2017; Mölg et al., 2018).

[Figure]

*Fig. 5. Overlaying of a 15m buffer (**a**) from the KGI-2020s glacier extent and a 30m buffer (**b**) from the supraglacial debris extent with MMD outlines on the base map of Sentinel-2 and Planet image.*

3. In Section 4, the authors should immediately state that due to the different approaches, data sources and methods cannot be compared without a high level of uncertainty and maybe, only qualitatively. Subsection 4.3 should be shortened and merged with the previous one.

Response: Thank you for your suggestions. We have stated this situation at the beginning of section 4.1. Subsection 4.3 has been shortened and merged into subsection 4.1 and subsection 4.2.

[revised manuscript text omitted]

6. In the data repository, the uncertainty statement is different from the one cited in the paper [±5.03% ≠ ±3.68%], please clarify.

Response: Thank you for pointing out the mistake. Due to the manuscript was revised twice according to the editor's suggestions before the interactive discussion, the uncertainty evaluation result changed. We neglected to synchronize the description of data assets. It has been revised now, and we will update all the information of the data assets when the revision of the manuscript is finished.

There are also minor comments as follows:

7. Subsection 2.5 is written in the wrong format;

Response: Thank you. We have revised it.

8. Line 485, please mention rockfalls in addition to avalanches;

Response: Thank you for your advice. The influence of rockfalls have been mentioned in this sentence.

*"Increased snow avalanche activity and **rockfalls** at high altitudes may have brought more debris to the glacier (Hewitt, 2005), thus…"*

9. when regression or correlation analyses are cited statistical significance (p value) and the correct parameter should be cited;

Response: Thank you for your suggestion. Correct and reasonable parameters (including correlation coefficient *r* and *p* value for evaluating significance) are updated and used for correlation and regression analysis in the revised manuscript. And the involved figures have been redrawn.

10. Maps should be improved by removing the north arrow and scale (the coordinates in the outline give the same information) or by placing them in the same area of the legend (i.e. unique white background).

Response: Thank you. According to your suggestion, we have improved all the figures (including supplementary figures) in the manuscript.

[revised manuscript text omitted]

---

## Editor Decision (ED1)

L140 ICESate --> ICESat

L258 is the correlation coefficient really equal to 1?

L293 32.5% --> 32.50%

L297 2,200 km2 --> 2200 km2

L300 55.3% --> 55.30%

Fig 6 I think that the most important information of this figure is the glacier area distribution and, second, the boundaries of Western, Central and Eastern (and maybe North and South) Karakoram. Yet, this information is the least evident, and it is not well explained in the caption. Rather, the most evident information is the division of the major sub-basins. Is this datum so relevant in this figure?
Fig 6 should be modified in other aspects.
i) in the caption, it should be added "(in dashed black line)" after the "main central ridgeline". You should highlight the central ridgeline better, because where it is superimposed is not well visible
ii) in the legend, the ridgeline NK_SK should be written explicitly
iii) the label of the southward arrow should be SK
iv) If I understand well, there are many sub-basins, which are labelled in black font. This should be explained in the caption. Not every sub-basin is labelled with its name. Why?
v) Karakoram limits should be highlighted more clearly
vi) add "river" after the label "Amu Darya"
vii) how can the Shyok basin be crossed by the central ridgeline NK-SK? Similarly, I see rivers that cross the central ridgeline. This is confusing
viii) Is Hotan Prefecture a city? Probably the city is Hotan only. Anyhow, the label is hidden behind the legend.
I suggest you to reconsider this figure and (strongly) modify it according to the very information that you want to show. Be also more exhaustive in the figure's caption

L327 74.4%--> 74.40%

L359 remove the tilde

Fig 7 Consider to split the figure into two or three figures, containing panels a-b, c-d and e-f. In any case, panels c-d and e-f should be on the same line (instead of the same column). Panel f should be on the left, in agreement with panel a, which shows debris-covered glaciers
Consider using a logarithmic scale in panel b, because it is difficult to distinguish between the different years. Alternatively, you may also add a subpanel that zooms on the interval 5000-6000 m
The x-label of panel a should be "debris-covered area (km2)"
Remove the colorbar in panels c-d. Moreover, is there any reason for displaying slices SE and E (in panel c) and SE (in panel SE) separated from the main circle? If not, you should display a solid circle.
In the caption
i) Plots showing glaciers' number and area by aspect (c and d) --> The pie charts show the distributions of glaciers' number (c) and area (d) according to the aspect
ii) Begin a new statement to describe panels e-f and remove "in addition".

L383 also --> conducted

L390 ~1 (0.997998) --> >0.99

L457 1,647.19 --> 1647.19

L463 ~70% --> Be more specific. In general, you provide very precise data with two decimal digits. I do not understand why in a few occasions you decide to be approximative

Fig 9, caption Hollow--> white

L478 What are the five sub-basins? The three basins are the coloured ones shown in fig 6 or EK, CK and WK?

L484 "with runoff moving towards peak water" is unclear

---

## Author Response (AR2)

Dear Niccolò Dematteis,

On behalf of all the co-authors, I would like to thank the reviewer, Anonymous Referee #1 and you again for your thoughtful and constructive suggestions which helped us to improve our manuscript. According to your suggestion, we carefully revised the manuscript. We have also stored KGI data in Geopackage Format and updated the data assets and description documents.

Please refer to the point-by-point reply (**next page**) or tracked manuscript version for details.

We hope that the revised manuscript will be more suitable for further processing, but we are still happy to consider further revisions, and we appreciate your efforts in our research.

Best regards,
Shiyin Liu,

Institute of International Rivers and Eco-security
Yunnan University
shiyin.liu@ynu.edu.cn

**NOTE**

Comments (Black font)

Authors Responses (Red font)

*Specific changes that were made in the manuscript (Blue italic)*

■ **Anonymous Referee #1**

The authors have substantially improved the manuscript, and after some minor corrections (see below), it will be suitable for publication.

1. Line 145, the URL should be removed as it is already reported in the following table. The sentence should be modified accordingly to the requested change.

Response: Thank you for your suggestion. We removed the URL and modified the sentence.

*"All ASTER GDEM v3 tiles were downloaded **from the Land Processes Distributed Active Archive Center**, and mosaicked in local Python GADL environment."*

2. For the sake of uniformity, please, add a white background to all legends, scalebar and north arrow inserted in maps.

Response: Thank you for your suggestion. In the latest revised manuscript, we make sure that the background of all legends, scalebar and north arrow are white.

3. Line 258, the p-value is missing.

Response: *p*-value has been added (p<0.001).

■ **Topical Editor, Niccolò Dematteis**

1. L140 ICESate --> ICESat

Response: Thank you. It has been revised.

2. L258 is the correlation coefficient really equal to 1?

Response: In fact, the correlation coefficient r is equal to 0.9989 (*p* < 0.001). With reference to the comments 11 (L390 ~1 (0.997998) --> >0.99), we changed it to 0.99.

3. L293 32.5% --> 32.50%

Response: Thank you. It has been revised.

4. L297 2,200 km$^2$ --> 2200 km$^2$

Response: Thank you. It has been revised.

5. L300 55.3% --> 55.30%

Response: Thank you. It has been revised.

6. Fig 6 I think that the most important information of this figure is the glacier area distribution and,

second, the boundaries of Western, Central and Eastern (and maybe North and South) Karakoram. Yet, this information is the least evident, and it is not well explained in the caption. Rather, the most evident information is the division of the major sub-basins. Is this datum so relevant in this figure? Fig 6 should be modified in other aspects.

i) in the caption, it should be added "(in dashed black line)" after the "main central ridgeline". You should highlight the central ridgeline better, because where it is superimposed is not well visible

ii) in the legend, the ridgeline NK_SK should be written explicitly

iii) the label of the southward arrow should be SK

iv) If I understand well, there are many sub-basins, which are labelled in black font. This should be explained in the caption. Not every sub-basin is labelled with its name. Why?

v) Karakoram limits should be highlighted more clearly

vi) add "river" after the label "Amu Darya"

vii) how can the Shyok basin be crossed by the central ridgeline NK-SK? Similarly, I see rivers that cross the central ridgeline. This is confusing

viii) Is Hotan Prefecture a city? Probably the city is Hotan only. Anyhow, the label is hidden behind the legend.

I suggest you to reconsider this figure and (strongly) modify it according to the very information that you want to show. Be also more exhaustive in the figure's caption

Response: Thank you very much for your advice. First of all, we apologize to you. Due to the revised version we submitted could not clearly distinguish between the new version and the old version of the figure 6, which led you to comment on the old version of the figure 6. But your suggestion still stands. We modified the figure 6 and its caption according to your suggestion. (NOTE: In the new revised version, we only keep the latest version of the figure.)

[Figure]

**Fig. 6**. *Spatial distribution of Karakoram glaciers. For regional comparison, the Karakoram mountains were divided into western, central and eastern Karakoram (WK, CK and EK) according to the Indus and Tarim sub-basins, and into Northern and Southern Karakoram based on the main central ridgeline (green line). The major sub-basin divisions (Wakhan, Gilgit-Hunza, Shigar, Sub-Tarim and Shyok) in the Karakoram mountains also are shown. Hollow circles and coloured grids represent the glacier area and glacier proportion on the 20 km × 20 km grid.*

7. L327 74.4%--> 74.40%

Response: Thank you. It has been revised.

8. L359 remove the tilde

Response: Thank you. It has been revised.

9. Fig 7 Consider to split the figure into two or three figures, containing panels a-b, c-d and e-f. In any case, panels c-d and e-f should be on the same line (instead of the same column). Panel f should be on the left, in agreement with panel a, which shows debris-covered glaciers

Consider using a logarithmic scale in panel b, because it is difficult to distinguish between the different years. Alternatively, you may also add a subpanel that zooms on the interval 5000-6000 m

The x-label of panel a should be "debris-covered area (km2)"

Remove the colorbar in panels c-d. Moreover, is there any reason for displaying slices SE and E (in panel c) and SE (in panel SE) separated from the main circle? If not, you should display a solid circle.

In the caption

i) Plots showing glaciers' number and area by aspect (c and d) --> The pie charts show the distributions of glaciers' number (c) and area (d) according to the aspect

ii) Begin a new statement to describe panels e-f and remove "in addition".

Response: Thank you for your advice. We split figure 7 into three figures, and then modified the problems in each figure according to your suggestion. Figures 7, 8, and 9 are as follows:

[Figure]

*Fig. 7 Altitudinal profiles of the glacier surface area at 50 m intervals for debris-covered ice (a) and all glaciers (b), showing variations from 1990 to 2020. The subpanel b₁ and b₂ inserted in figure (b) correspond to zooms on the altitude interval 5000~6000 m a.s.l. and 3500~4000 m a.s.l..*

[Figure]

*Fig. 8* *The pie charts show the distributions of glaciers' number (a) and area (b) according to the aspect.*

[Figure]

*Fig.9* *Glacier surface slope versus latitude for debris-covered sections (a) and all glaciers (b).*

10. L383 also --> conducted

Response: Thank you. It has been revised.

11. L390 ~1 (0.997998) --> >0.99

Response: Thank you. It has been revised.

12. L457 1,647.19 --> 1647.19

Response: Thank you. It has been revised.

13. L463 ~70% --> Be more specific. In general, you provide very precise data with two decimal digits. I do not understand why in a few occasions you decide to be approximative

Response: The specific proportion is 71.43% (45/63). The number of surge-type glaciers from the most recent inventory of surge-type glaciers (Guillet et al., 2022), and the number of advancing glaciers from KGI-2020s data.

14. Fig 9, caption Hollow--> white

Response: Thank you. It has been revised.

15. L478 What are the five sub-basins? The three basins are the coloured ones shown in fig 6 or EK, CK

and WK?

Response: The five sub-basins are: Wakhan, Gilgit-Hunza, Shigar, Sub-Tarim and Shyok. Now, they are shown in figure 6, also list in Table S6.

16. L484 "with runoff moving towards peak water" is unclear

Response: Thanks. Now, we changed the statement to *"On the whole, the median elevation of the Karakoram glaciers showed an increasing trend during 1990-2020, indicating that glacier melting likely is becoming more intense, **the annual glacier runoff is moving towards a maximum (peak meltwater)**"*.

**References**

Guillet, G., King, O., Lv, M., Ghuffar, S., Benn, D., Quincey, D., and Bolch, T.: A regionally resolved inventory of High Mountain Asia surge-type glaciers, derived from a multi-factor remote sensing approach, The Cryosphere, 16, 603-623, 10.5194/tc-16-603-2022, 2022.